# Systematic Identification of the Optimal Housekeeping Genes for Accurate Transcriptomic and Proteomic Profiling of Tissues following Complex Traumatic Injury

**DOI:** 10.3390/mps6020022

**Published:** 2023-02-23

**Authors:** Andrea H. Dragon, Cassie J. Rowe, Alisha M. Rhodes, Olivia L. Pak, Thomas A. Davis, Elsa Ronzier

**Affiliations:** 1Cell Biology and Regenerative Medicine Program, Department of Surgery, Uniformed Services University of the Health Sciences, 4301 Jones Bridge Road, Bethesda, MD 2081, USA; 2Henry M. Jackson Foundation for the Advancement of Military Medicine, 6720A Rockledge Drive, Bethesda, MD 20817, USA

**Keywords:** trauma, gene expression, housekeeping gene (HKG), reference gene normalization, gene expression stability, immunoblot, real-time quantitative polymerase chain reaction (RT-qPCR), secondary organs, rat

## Abstract

Trauma triggers critical molecular and cellular signaling cascades that drive biological outcomes and recovery. Variations in the gene expression of common endogenous reference housekeeping genes (HKGs) used in data normalization differ between tissue types and pathological states. Systematically, we investigated the gene stability of nine HKGs (*Actb*, *B2m*, *Gapdh*, *Hprt1*, *Pgk1*, *Rplp0*, *Rplp2*, *Tbp*, and *Tfrc*) from tissues prone to remote organ dysfunction (lung, liver, kidney, and muscle) following extremity trauma. Computational algorithms (geNorm, Normfinder, ΔCt, BestKeeper, RefFinder) were applied to estimate the expression stability of each HKG or combinations of them, within and between tissues, under both steady-state and systemic inflammatory conditions. *Rplp2* was ranked as the most suitable in the healthy and injured lung, kidney, and skeletal muscle, whereas *Rplp2* and either *Hprt1 or Pgk1* were the most suitable in the healthy and injured liver, respectively. However, the geometric mean of the three most stable genes was deemed the most stable internal reference control. *Actb* and *Tbp* were the least stable in normal tissues, whereas *Gapdh* and *Tbp* were the least stable across all tissues post-trauma. Ct values correlated poorly with the translation from mRNA to protein. Our results provide a valuable resource for the accurate normalization of gene expression in trauma-related experiments.

## 1. Introduction

Trauma is the leading cause of morbidity and mortality worldwide [1,2,3,4]. Injury is primarily the result of transportation-related events, firearm discharges, industrial workplace injuries, and natural disasters [5,6,7,8,9,10], wherein traffic-related injuries outweigh the others, particularly in low- and middle-income countries [2]. The patterns of injury among critically injured combat casualties present additional trauma management challenges with regard to injury-related sequelae and medical interventions. Understanding the pathophysiology of various forms of severe trauma and the underlying molecular mechanisms are based largely on reliable animal model studies and military outcome data [11,12].

Trauma activates the cellular and molecular immune response at the site of injury. Distinct local and remote molecular perturbations in gene expression detected early after injury are critical in regulating physiological processes throughout the body, including tissue-repair resolution. These molecular changes precede the occurrence of subsequent complications such as inflammation, systemic inflammatory response syndrome (SIRS), multiple organ dysfunction syndrome (MODS), and severe infection (sepsis) [13,14,15]. Transcriptomic profiling is a powerful comprehensive approach to investigating early molecular pathways and biological processes that trigger and regulate physiological and pathological conditions following trauma. This approach aids in the identification of therapeutic targets, as well as gene expression patterns, in response to treatment.

Real-time quantitative polymerase chain reaction (RT-qPCR) is an accurate method for gene expression analyses; however, the accuracy of the normalized results is affected by the expression stability of internal reference control genes [16]. Low- and high-density RT-qPCR microarrays are routinely used platform techniques to rapidly and reliably quantitate a large number of mRNA molecules from multiple samples in parallel [17]. The choice of an appropriate internal reference control (stable endogenous unregulated reference housekeeping gene(s) (HKGs)) for data normalization is a critical step in identifying relevant changes in gene regulation processes, which may be involved in biological and pathological processes in the same sample. A comprehensive literature analysis of gene expression studies published in 1999 showed *Actb*, *Gapdh*, and *18S* were conventionally used 90% of the time as single reference control genes [18]. To achieve accurate target gene normalization, expression levels of reference HKGs should remain stable between the cells of different tissues and under different experimental conditions. If the experimental condition or pathology results in variable directional shifts in the expression of a particular HKG, the subsequent target gene normalization will lead to erroneous results [19,20]. Similarly, normalizing target protein expression to a stable housekeeping protein is critical for reliable and accurate quantitation of mRNA translation, which results in protein production under both normal physiological conditions and following trauma [21,22]. Therefore, the appropriate selection of cell-tissue-specific reference genes as internal reference controls must be experimentally validated for RT-qPCR data normalization in order to guarantee the correct analysis of observations and the quality of results.

Major differences in the levels of expression and stability of common endogenous reference HKGs have been reported between numerous tissue types and pathological states [23,24,25]. Traditional HKGs such as *Gapdh* and *Actb* are frequently utilized as internal reference controls in both genomic and proteomic studies, however, in many cases, are inappropriately selected due to their expression variability between experimental conditions or pathologies such as inflammatory diseases and cancers [26,27]. Furthermore, little is known about HKG expression and stability in relation to the steady-state and the early trauma-induced inflammatory and wound healing immune response. A literature search for wound-healing experiments published in the Journal of Wound Repair and Regeneration (January 2008–August 2009) reports that *Actb*, *Gapdh*, *18S*, and *B2m* were the most frequently used housekeeping genes for RT-qPCR data normalization in human, mouse, and pig studies, independent of validation studies, to confirm differential gene expression [28]. Others have reported high variability in *Gapdh* and *Actb* expression across experimental timepoints in animal models of trauma-induced inflammation involving the sciatic nerve lung, brain, and skin, in addition to sepsis [29,30,31,32,33,34,35]. Since transcripts of HKGs can vary considerably in tissues under different experimental conditions, it is imperative to validate the expression stability of reference HKGs following acute orthopedic trauma.

In the present study, we validated and identified appropriate control HKGs for genomic and proteomic data normalization in a variety of tissues in both the normal steady-state and pathological state following acute injury in a rat model of blast-associated combat-related lower limb trauma (Figure 1). By employing RT-qPCR and immunoblot techniques and computational analyses, we evaluated the variability in gene and protein expression among nine common HKGs: *Actb*, *B2m*, *Gapdh*, *Hprt1*, *Pgk1*, *Rplp0*, *Rplp2*, *Tbp*, and *Tfrc*. We used the RT-qPCR results and applied five independent and rigorous statistical algorithms to compare and rank HKG stability (i.e., RefFinder, Normfinder, BestKeeper, and geNorm, as well as the comparative Delta-Ct method [19,30,36,37,38,39]). In addition, we examined the tissue-level protein-coded expression of each HKG in the steady-state by immunoblots. Temporal changes in HKG expression between and within tissue types, both prior to and after injury, were investigated to define the most stable and comprehensive reference HKGs. This study highlights the importance of reference gene stability analysis and demonstrates how the improper choice of internal reference control HKGs can significantly impact the robustness of normalization and lead to data misinterpretation. Furthermore, we provide validated reference gene candidates for rat lung, kidney, liver, and skeletal muscle prior to and after acute extremity trauma.

## 2. Materials and Methods

### 2.1. Animals

Eleven- to twelve-week-old adult male Sprague–Dawley rats (n = 35) were subjected to an established model of extremity trauma that incorporates some of the critical elements associated with combat injury, specifically systemic blast injury (120 kPa; previously documented to result in mild traumatic brain injury [TBI]) [40]; femur fracture with muscle crush injury; 3 hours (h) of tourniquet-induced ischemia, followed by a transfemoral hindlimb amputation after a 1 h limb reperfusion, as previously described [41,42]. Lung, kidney, liver, and injured skeletal muscle were collected after euthanasia at 6, 24, and 168 h (7 days) post-injury (n = 7 rats/timepoint). Tissues collected from healthy age-matched naïve uninjured rats served as the controls (n = 14 animals). Tissue samples were immediately flash-frozen in liquid nitrogen and stored at −80 °C.

### 2.2. RNA Isolation, cDNA Synthesis, and Real-Time Quantitative Polymerase Chain Reaction (RT-qPCR) Analysis

Frozen tissue samples (400 mg) were transferred into 2 mL ceramic bead tubes (VWR, Randor, PA, USA) containing 500 µL of Qiazol lysis reagent (Qiagen, Valencia, CA, USA). The tubes were placed in a Mini Bead Mill Homogenizer (VWR) and the samples were homogenized for 120 seconds (s) at 4 m/s. The tubes were further centrifuged at 12,000 rcf at 4 °C for 15 minutes (min). Supernatants were collected and total RNA was isolated. RNA isolation was conducted using the RNeasy mini and DNase I kits (Qiagen) following the manufacturer’s specifications. The concentration of total mRNA (ng/μL) and the purity (A260/A280 ratio) were determined using a Nanodrop (ThermoFisher Scientific, Waltham, MA, USA). RNA preparations with an A260/A280 purity ratio value between 1.8 and 2.2 were considered acceptable. First-strand cDNA was generated from 600 ng of total RNA using the iScript Advanced cDNA synthesis kit (Bio-Rad, Hercules, CA, USA) according to the manufacturer’s specifications. We selected nine common reference housekeeping genes based on a targeted literature search of the PubMed database (2013–2023) for qPCR data normalization studies (Appendix A) across diverse spectrums of tissues, disease models, and animal species. For the most common reference genes reported in the literature, it was not always clear on what basis these control genes were chosen, and for those used for gene expression normalization, which HKGs were the most appropriate. It is important to note that very few of these reports involved critical validation studies. The specifics of the nine candidate HKGs tested and the primer/probe sets used in this study are presented in Table 1. qPCR reactions containing 10 ng of cDNA were performed with SsoAdvanced Universal SYBR Green Supermix (Bio-Rad; cat# 172-5275) with ROX normalization using the QuantStudio 7 Pro Real-Time PCR system (Applied Biosystems, Foster City, CA, USA). The fast-run PCR program consisted of an initial denaturation step at 95 °C for 2 min, followed by an amplification step of 40 cycles at 95 °C for 5 s and 60 °C for 30 s. Cycle threshold (Ct) values of the 9 candidate reference genes were determined via the ThermoFisher Connect Cloud-based Software Platform. The melting curve protocol consisted of 15 s at 95 °C and 10 s at each 0.3 °C increment step from 60° to 95°.

### 2.3. PCR Amplicon Validation

Primer validation measurements were determined by analysis of the melting curves of the individual amplicons. The melting curves were taken from a single assay with four replicates for each target HKG. PCR amplification products were verified using gel electrophoresis to determine amplicon length. The gel was made with 2% agarose (Millipore Sigma, Burlington, MA, USA) dissolved in TAE (Tris-Acetate Ethylenediaminetetraacetic acid [EDTA]; ThermoFisher Scientific) supplemented with 1:10,000 SYBR dye (ThermoFisher Scientific). A low-molecular-weight DNA ladder (New England Bio lab, Ipswich, MA, USA) was run in parallel with the qPCR products at 120 V for 45 min. The PCR products were visualized using a ChemiDoc Imaging System (Bio-Rad).

### 2.4. Protein Lysate Preparation and Protein Expression Analysis through Western Blot

Samples were homogenized in radioimmunoprecipitation assay (RIPA) lysis reagent (ThermoFisher Scientific) supplemented with 1% of protease and a phosphatase inhibitor cocktail (Millipore Sigma). Protein concentration was measured using the Pierce bicinchoninic acid (BCA) Protein Assay Kit (ThermoFisher Scientific). Twenty microliter samples containing NuPage LDS Sample Buffer (4×; ThermoFisher Scientific), sample reducing agent (10×; Fisher Scientific, Hampton, NH, USA), and 15 µg of protein were separated by 4 to 12%, Bis-Tris acrylamide/polyacrylamide gel, 1.0 mm on a Mini SDS-PAGE Protein Gel (ThermoFisher Scientific) electrophoresis device and subsequently transferred to a nitrocellulose membrane (0.2 µm, ThermoFisher Scientific) by electrophoresis. Ponceau S staining was conducted to document the total protein amount loaded per lane. The membrane was stained with Ponceau S Solution (Abcam, Cambridge, UK, cat# AB270042) for 5 min at room temperature and washed briefly twice with distilled water to remove the background. Imaging of the membrane was conducted using a colorimetric imager (ChemiDoc Imager, Bio-Rad). The membrane was blocked for one hour in a 5% nonfat milk or 5% BSA solution and then probed overnight at 4 °C with primary antibodies against rat *ACTB*, *B2M*, *GAPDH*, *HPRT1*, *PGK1*, *RPLP0*, *RPLP2*, *TBP*, and *TFRC* (Appendix A). The membranes were washed three times and incubated with horseradish peroxidase-conjugated secondary antibodies (1:10,000) for one hour at room temperature. Protein bands were visualized using an Immobilon Western chemiluminescent HRP Substrate kit (Millipore Sigma, Burlington, MA, USA) and chemiluminescence imager (ChemiDoc Imager, Bio-Rad).

### 2.5. Data Analysis

#### 2.5.1. Gene Expression and Stability Analysis

The expression stability of candidate HKGs across the various tissues and experimental conditions was analyzed using the following five mathematical algorithms: RefFinder [17] (https://heartcure.com.au; accessed on 11 March 2022), Normfinder [18], BestKeeper [19], and geNorm [20], and the comparative Delta-Ct method [21]. All values are expressed as the mean ± standard deviation of the mean, except where indicated. Outliers were defined as Ct values greater than two standard deviations from the mean and were removed from the data. Coefficients of gene expression variation (CV) were calculated as the standard deviation normalized to the expression mean. The geometric means of the Ct values were calculated using combinations of the top-three most stable HKGs identified by the ReFinder comprehensive ranking algorithm. The relative change in expression of the target genes was analyzed using the 2^−ΔΔCt^ method [42,57,58,59,60].

#### 2.5.2. Comprehensive Ranking Assessment

Eight different parameters were used to rank the nine HKGs, Ct value, Ct value standard deviation, and gene stability, taking into account the five statistical algorithms (RefFinder [17] (https://heartcure.com.au; accessed on 11 March 2022), Normfinder [18], BestKeeper [19], and geNorm [20], and the comparative Delta-Ct method [21]) and the protein expression level. The sums of the ranking for each individual HKG were calculated. The smallest sum represents the HKG with the best fit for each organ, whereas the highest calculated sum represents the most unsuitable.

#### 2.5.3. Statistics

All statistics were calculated from the mean of the independent experiments using Prism version 9.4.1 (GraphPad Software, San Diego, CA, USA) using unpaired *t*-tests with α = 0.05. Statistical significance was defined as * *p* ≤ 0.05; ** *p* ≤ 0.01, *** *p* ≤ 0.001, and **** *p* ≤ 0.0001.

## 3. Results

### 3.1. Assessment of Primer Specificity

We assessed the gene-specific amplification of the nine rat-specific HKG RT-qPCR primers by analyzing the melting curves and qPCR amplicons. The melting curve analyses revealed single peaks for each primer pair, indicating the specificity of the RT-qPCR reactions (Figure 2A). This was confirmed by analyzing the final PCR products on the 2% agarose gel, which showed the presence of a single product at the expected amplicon size (Figure 2B and Table 1). These findings indicate that no primer dimers or nonspecific amplification products were formed.

### 3.2. Evaluation of Gene Transcripts and Protein Expression Variability of HKGs in Healthy Tissue

We began our systematic ranking by comparing the raw Ct values of the nine HKGs within each organ (Figure 3A and Appendix A). The nine HKGs displayed a wide expression range, with Ct values ranging between 19 and 30 across all four tissues tested. *Rplp0*, *B2m*, *Pgk1*, *Rplp2*, and *Actb* were highly expressed in the naïve lung (Ct_mean_: 20.50–23.77), whereas abundant *Rplp0*, *Pgk1*, *Gapdh*, *Rplp2*, and *B2m* expression levels (Ct_mean_: 19.18–24.64) were observed in the naïve kidney, liver, and skeletal muscle. *Tbp* and *Tfrc* showed the least expression in the healthy lung, liver, and kidney (Ct _mean_: 27.71–30.26). *Actb* and *Tbp* had low transcript expression (Ct _mean_: 28.85–30.92) in the skeletal muscle (Figure 3 and Appendix A). Of the nine HKGs, *Pgk1* had the least variation across all the organs (CV_mean_: 0.034). The muscle had the most variable expression level across all nine HKGs (CV_mean_: 0.116), whereas all other tissues were comparable.

We next confirmed equal immunoblot sample loading by Ponceau S staining (Appendix A) and then the HKG mRNA transcripts and protein across all four tissue types in the steady-state by immunoblot analysis (Figure 3C and Appendix A). We then ranked the candidate HKGs based on protein expression for each tissue type (Figure 3D). Across the four tissues tested, we found significant differences in the ratios between protein and mRNA transcripts, which were mainly determined by translation and protein turnover/degradation [61]. *Actb* and *B2m* were the most abundant housekeeping proteins in the healthy lung, liver, and kidney, with reduced abundance in the skeletal muscle. Conversely, *Gapdh* was strongly detectable in the skeletal muscle, with modest production in the lung, liver, and kidney (Figure 3C). We were able to detect the low-abundance proteins *B2m* and *Actb* in the skeletal muscle and *Gapdh* in the kidney by increasing the amount of total protein loaded (Appendix A). Immunoblot analysis of the liver depicted the presence of seven out of nine housekeeping protein (HKP) candidates. Compared to other HKPs, TBP, and TFRC displayed reduced intensity across all tissues (Figure 3C).

Based on both the gene and protein expression results depicted in Figure 3A–D, we summed the individual rankings of each of the eight different parameters to determine the most suitable HKG for each tissue in the naïve steady-state. The resultant ranking (max score: 64) indicated the suitability, with the lowest sum indicating the best suited. Unequivocally, *Rplp2* was the most suitable endogenous reference for the healthy naïve lung, liver, and skeletal muscle, whereas *Rplp2* and *Hprt1* ranked equally for the kidney (Figure 3E).

### 3.3. Expression of Reference HKG Genes following Trauma

The aforementioned stability ranking scheme described for healthy tissue samples was applied following trauma-induced stimuli (Figure 4). *Rplp0*, *Pgk1*, *B2m*, *Rplp2*, and *Gapdh* were highly expressed among naïve and injured tissues, whereas *Tbp* and *Tfrc* were negligibly expressed (Figure 4A and Appendix A). We observed the largest variation in expression in injured lung and liver tissue for all nine HKGs compared to the naïve. The greatest variability was detected in the kidney, as *B2m* (Ct: 23.47 ± 2.19; CV: 0.093) and *Actb* (Ct: 26.93 ± 4.75; CV: 0.176) had the most variable expression levels under traumatic injury conditions compared to the naïve. Compared to other tissues, CVs obtained from the injured skeletal muscle tissues were equal to or smaller than the CVs obtained for the naïve condition (Figure 4A and Appendix A), indicating that the mechanical injury to the hindlimb and amputation does not result in the variability of HKG expression. Based on the computational algorithms alone, the stability rankings for each tissue type in the injured state were the most consistent in the lung and the least consistent in the liver (Figure 4B) compared to the stability rankings of the tissues in the naïve steady-state. Consistent with the gene expression results from the naïve steady-state, *Rplp2* was the most stable reference gene for the injured lung, kidney, and muscle. However, there were observed differences in the stability ranking in the injured liver compared to the naïve steady-state, as *Hprt1* ranked as the most stable in the injured liver (Figure 4B), whereas *Hprt1* previously ranked third in the naïve steady-state. The rankings were the least consistent in the liver and the most consistent within the lung (Figure 4B).

Lastly, we combined the naïve and injured samples and ranked the HKGs using our multi-parameter ranking method (Figure 4C) in order to determine the most suitable housekeeping gene for the entire data set. Across all experimental conditions (naïve and injured states), there was no universal HKG that ranked as most stable among all four tissue types; however, *Rplp2* was recognized as the highest-ranked suitable candidate gene in the lung, kidney and skeletal muscle, whereas *Hprt1* and *Pgk1* were equally ranked for the liver (Figure 4C). We observed that the liver had the highest overall disparity between the naïve and traumatic states, as seen in the raw mean Ct values and gene stability rankings.

### 3.4. Validation of the Tissue-Specific HKG Selection via Calculated Expression Levels of Known Inflammatory Biomarkers after Severe Trauma

Dysregulated hyperinflammation involving a myriad of systemically released pro- and anti-inflammatory cytokines (“cytokine storm”) through the activation of the immune system can provoke many secondary injuries during the early recovery phase, including ischemia-reperfusion injury, multi-organ dysfunction syndrome (MODS), organ failure, and sepsis. Interleukin 6 (*Il6*), tumor necrosis factor (*Tnf*), and myeloperoxidase (*Mpo*) are just a few of the key factors/targets that play key roles in this response [62,63,64,65,66]. To demonstrate the profound impact of appropriate reference gene selection for calculating the relative gene expression of a specific target gene of interest, we analyzed the pro-inflammatory mediators *Mpo*, *Tnf*, and *Il6* at 6 and 24 h post-injury using three normalization strategies: the identified most stable HKG, the least stable HKG, and a combination of the three most stable genes at each timepoint. First, we illustrate the raw mean Ct value variability of the most and least stable HKGs across the tissues and experimental conditions (naïve, 6 h, and 24 h post-injury; Figure 5A). The relative expression of the target genes examined was dependent on the normalization strategy used (Figure 5B). In general, the expression levels were, in many cases, strongly overestimated when using unstable HKGs for normalization. This was most apparent in the liver, particularly for calculating *Mpo* expression 6 h post-injury, where there was a 2-log expression difference using the least or most stable HKG(s).

The expression stability of either a single gene or a combination of the two or three most stable HKGs was then analyzed using the RefFinder comprehensive analysis, taking into consideration the five computational algorithms (Appendix A). In each tissue type, the geometric mean of the top-three genes improved the stability of the normalization factor compared to each gene individually. Furthermore, when used to compute the expression levels of known inflammatory biomarkers following traumatic injury, the calculated fold-change was comparable within timepoint groups and the intersample variability was greatly reduced. From this analysis, it was clear that the experimental results obtained using the least unstable HKG differed greatly from those using a validated reference gene or a combination of stable HKGs, resulting in major erroneous directional shifts and biologically conflicting results of significant magnitude.

## 4. Discussion

Gene expression analysis plays an instrumental role in our understanding of diseases and treatment effects. The reproducibility and reliability of transcriptomic results obtained from real-time quantitative polymerase chain reaction (RT-qPCR) are dependent upon the sample quality, primer specificity, and selection of ubiquitously expressed reference control genes that are unaffected at the transcriptomic level by the experimental conditions [16,18,19,20,67,68,69,70,71]. In general, the selection of internal reference housekeeping genes (HKGs) for RT-qPCR gene expression data normalization is largely based on commonly used HKGs reported in the literature; rarely used HKGs are selected by conducting a gene stability analysis based on all experimental conditions. To the best of our knowledge, this is the first study to systematically compare the robustness and reliability of several common HKGs as suitable endogenous RT-qPCR reference controls within four tissue types prior to and after blast-related extremity trauma. The main findings of this study are as follows: (1) of the profiled HKGs, *Rplp2* mRNA expression had the least variance and was ranked as the most stable HKG in both healthy and injured lung, kidney, and skeletal muscle tissues; (2) *Rplp2* and either *Hprt1 or Pgk1* mRNA expression had the least variance and they were ranked as the most stable in the healthy and injured liver, respectively; (3) severe injury led to highly variable expression in all tissues (lung, liver, kidney), with the exception of the skeletal muscle; (4) the gene stability ranking results obtained using the comparative Delta-Ct method, GeNorm, and NormFinder were consistent, whereas BestKeeper ranked *Pgk1* as the most stable HKG among the nine candidates across all four organs and all experimental conditions; (5) overall *Actb* and *Tbp* were the least reliable HKGs in normal tissues, whereas *Gapdh* and *Tbp* were consistently unreliable across all tissues post-trauma; and (6) the Ct values of the nine HKGs in all four tissues correlated poorly with the translation of genetic information from mRNA to protein abundance. These findings are consistent with recent reports that commonly used HKGs, such as *Gapdh*, *B2m*, and *Actb*, have critical limitations in certain physiological states, models of injury, inflammatory diseases, cancers, wound healing, sepsis, and burn injuries [14,25,26,27,28,29,31].

We showed that through the use of multiple known methods for HKG validation, the results from each test varied. Our HKG stability rankings were determined using five algorithms (geNorm, NormFinder, BestKeeper, and the ΔCt method), as well as the comprehensive algorithm RefFinder, which considered the individual rankings and provided a comprehensive stability ranking. Using RefFinder, we demonstrate that *Rplp2* was the best normalizing gene across all tissues and study conditions, with the exception of the injured liver, where *Hprt1* was ranked as the most stable. Compared to all other algorithms, the ranking according to the BestKeeper coefficient of correlation showed a different output, identifying *Pgk1* as the most stably expressed gene in healthy and injured lung, liver, and muscle tissues. These discrepancies, particularly between BestKeeper and the other algorithms, are attributable to the differences in the calculation strategies and have been reported previously [38]. BestKeeper calculates the stability of the candidate genes based on the SD of their Ct values [38]. Despite these slight differences, the overall trends of the five methods were well correlated in the stability rankings. However, caution is recommended when using a single algorithm as a benchmark.

We speculate that the noted variability in the expression stability profiles for some of the HKGs in tested tissues post-trauma may be a result of changes in cellular composition resulting from immune cell recruitment, activation, and infiltration in response to cascades of metabolic and inflammatory perturbations. We have observed that skeletal muscle injury initiates the production and release of proinflammatory chemokines/cytokines which serve as recruitment-activation signals for neutrophils. At the site of tissue damage, infiltrating neutrophils promote excessive release of reactive oxygen species (ROS), proteinases, proinflammatory cytokines and chemokines [42,72]. Out of all the HKGs tested, the ribosomal gene *Rplp2*, which is involved in cell metabolism and regulation [73], showed the best stability ranking in all tissues pre- and post-injury, except for the kidney, and seems to be a safe choice for accurate expression rate determinations for tissue target genes with relatively high endogenous expression levels under the experimental conditions assessed in this study. HKGs widely used to standardize both mRNA transcript and protein levels in tissues, such as *Gapdh*, *Actb*, and *Tbp*, which play major regulatory roles in cellular activation, proliferation, and differentiation activities [43,44,45,48,49,55,74], were ranked as the least stable in both states. The mRNA translation levels were similarly highly variable across the four studied healthy tissues. Indeed, the comparison of the protein transcript levels of the nine HKG candidates showed a clear difference between the four evaluated tissues. Surprisingly, *B2m* and *Actb* were not highly translated into protein in the muscle tissue samples in comparison to the lung, liver, and kidney and, conversely, GAPDH protein expression was greater in the muscle tissue in comparison with the three other tissue types evaluated. In our trauma model, these three specific HKGs have critical limitations and are not suitable reference controls for trauma-related experiments, as they exhibit expression variability at both the transcriptional and protein levels. Based on all the data obtained, it can be concluded that the suitability of HKGs cannot be assumed, and it is necessary to identify and validate unique HKGs for each tissue type and specific research question being investigated. The absence of HKG validation likely contributes to the lack of reproducibility and generation of disparate results amongst replicate studies and between similar studies.

Importantly, we demonstrated that the inappropriate selection of reference HKGs can lead to bias and substantial interpretation errors, resulting in the under- or overestimation of target gene expression levels computed from the same dataset. The gene expression of a target gene of interest is usually expressed in relation to one or multiple reference genes; therefore, the accurate normalization of the data is of critical importance to generating comparable results within and across studies. Comparing the *Il6*, *Mpo*, and *Tnfa* expression data normalized using the most stable single or combination (geometric mean) of HKG(s) identified by RefFinder against the least stable HKG, we calculated profound overestimation differences in the final results. In general, using the most stable HKG versus a combination of the top-three stable HKGs for target gene normalization produced similar results, but this highlights the importance of validating the stability of reference genes before normalization.

There were several limitations to this study. First, our sample size was relatively small, albeit we detected strong transcriptomic HKG signatures. Second, we evaluated only the stability ranking of nine common HKGs among thousands of candidate reference genes; therefore, it is likely that there are other optimal reference genes or combinations that may be better suited to conduct precise quantitative analysis [75]. Lastly, the data from this study is likely specific to the normal steady-state and blast-related extremity injury conditions used in this study’s male Sprague–Dawley rat model. Therefore, control HKG selection must be validated for each experimental model, cell type, and tissue at early, intermediate, and later timepoints throughout the post-injury time course.

The innate immune response to critical injury is complex and vital to survival. Often an excessive, inappropriate, or dysregulated inflammatory immune response leads to widespread triggers of cellular activation, innate cell infiltration, cellular injury, and systemic inflammation, which sets the trajectory for multiple organ dysfunction pathophysiology involving the lung, kidney, liver, gut, and heart [76], the mechanism of which requires further study. Due to pivotal factors and evolving conditions within tissues following trauma, our findings suggest that careful validation of HKG amplification in each tissue at each timepoint is required to determine the accurate quantitative changes in target gene expression. We demonstrated critical differences with respect to HKG expression between tissues prior to and after trauma and showed that no single HKG is stable in every tissue under all conditions. Compared with previously published procedures for identifying suitable normalization genes using gene expression, the present approach takes into account tissue protein expression. Our stability findings correlate with the biological regulatory functions of these genes in both the steady-state and metabolically stressed inflammatory active tissues following trauma-induced conditions. Together, these findings highlight the importance of reference gene stability analysis and show how improper reference genes can lead to significant dimensions of data misinterpretation, bias, and false conclusions, especially when the results are not validated by complementary protein quantification data. These data provide validated reference gene candidates for the accurate assessment of differential target gene expressions in future studies.

## Figures and Tables

**Figure 1 mps-06-00022-f001:**
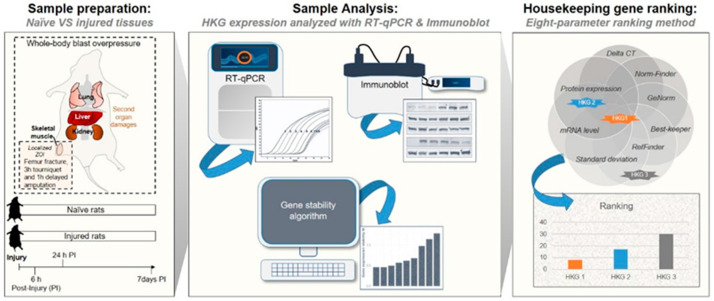
Method and protocol flow design.

**Figure 2 mps-06-00022-f002:**
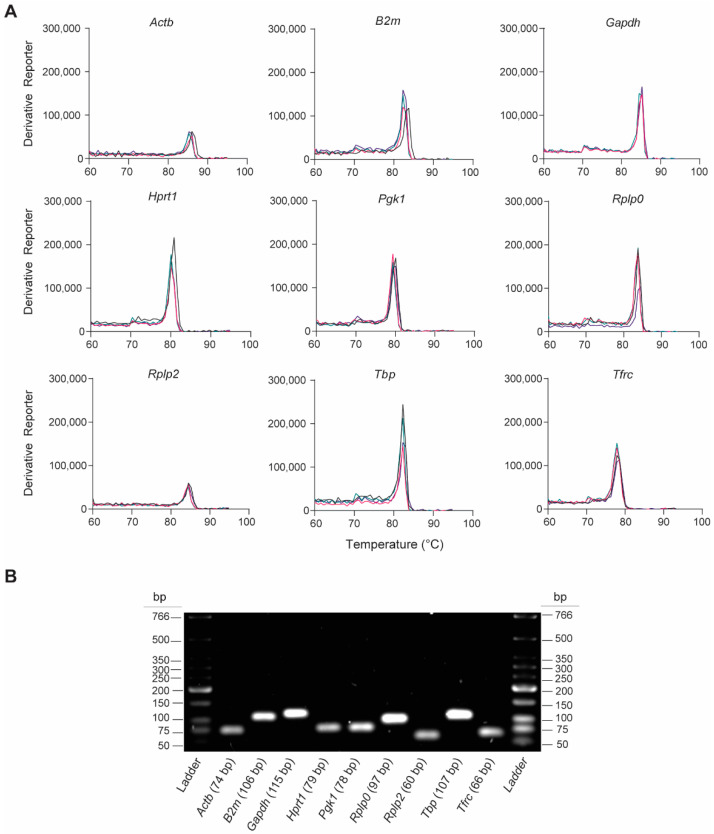
Validation of RT-qPCR primer specificity for the nine candidate rat-specific HKGs. (**A**) Representative melting curves obtained from qPCRs reactions, which show the peak of DNA dissociation for each HKG in lung tissue 6 h post-injury; n = 4 individual melting curves. (**B**) Representative image of a 2% agarose gel electrophoresis containing PCR amplicon products (7.5 µL/lane) obtained after RT-qPCR.

**Figure 3 mps-06-00022-f003:**
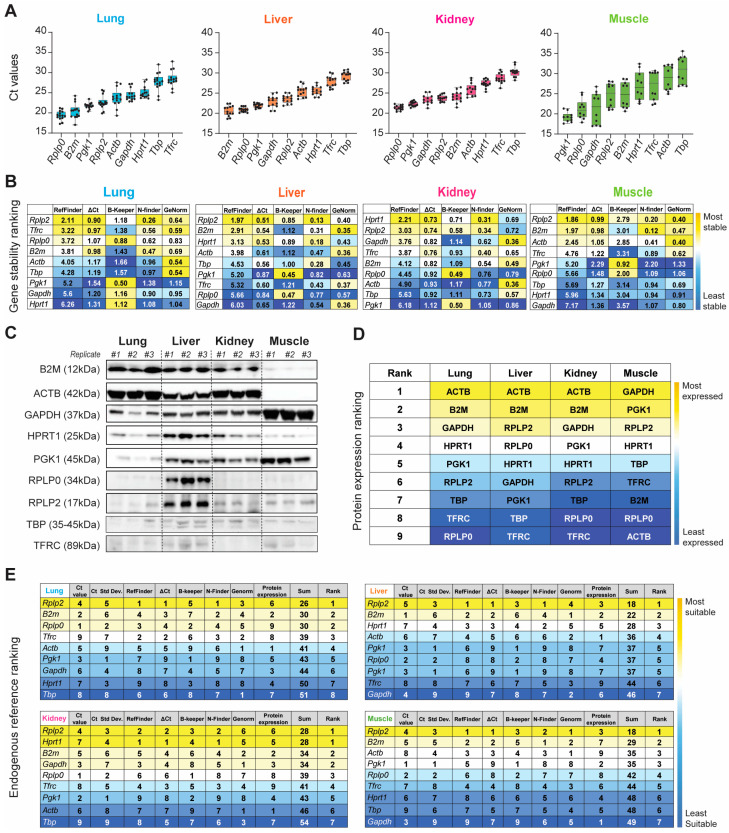
Evaluation of gene expression, stability, and translation of the nine HKGs between the lung, liver, kidney, and muscle tissue in naïve rats. (**A**) Ascending order distribution of the raw mean Ct values of the nine HKGs (n ≥ 10). Error bars represent the standard deviation of the mean. (**B**) Heatmaps representing the gene stability rankings obtained with RefFinder, Delta Ct (ΔCt), BestKeeper (B-Keeper), NormFinder (N-finder), and GeNorm for the nine HKGs. The color scale represents the most stable gene (yellow) to the least stable gene (blue; n ≥ 10 rats). (**C**) Representative Immunoblots against the 9 HKGs obtained with 15 µg of total protein loaded. (n = 3 biological replicates indicated as #1, #2, and #3). (**D**) Protein expression ranking for the nine HKGs. The color scale represents the most (yellow) to the least (blue) abundant protein expression. Results extrapolated from immunoblots presented in naïve. (**E**) Ranking of the nine HKGs according to their Ct value, Ct value standard deviation, gene stability from the five algorithms, and protein expression level. The color scale represents the most (yellow) to the least (blue) suitable endogenous reference.

**Figure 4 mps-06-00022-f004:**
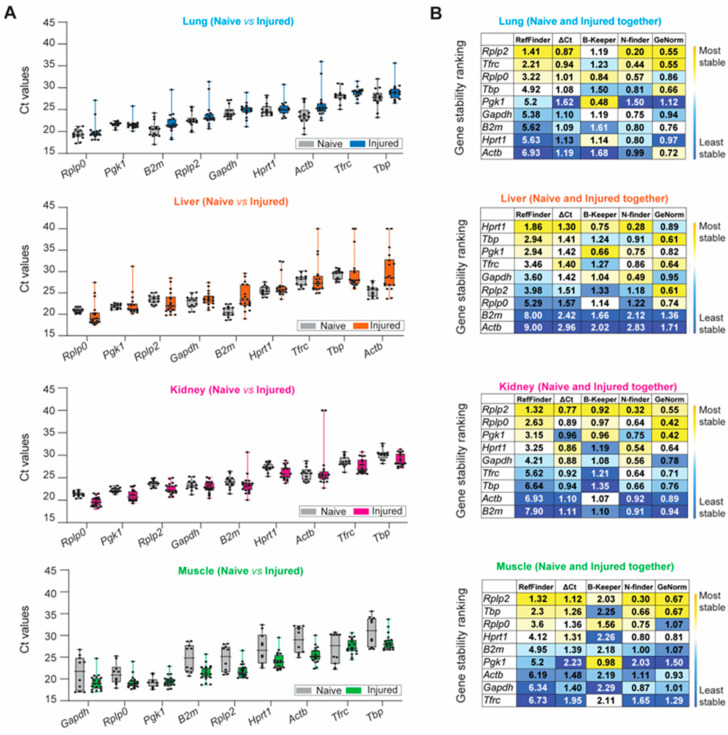
Identification of the most suitable endogenous reference in different experimental conditions (naïve and injured tissue) of the lung, liver, kidney, and muscle. (**A**) Ascending order distribution of the Ct values for the nine HKGs in the steady-state (naïve, grey) and injured tissues (injured, colored) (n ≥ 10 rats). Error bars represent the standard deviation of the mean. (**B**) Heatmaps representing the gene stability rankings obtained with RefFinder, Delta Ct (ΔCt), BestKeeper (B-Keeper), NormFinder (N-finder), and GeNorm for the nine HKGs by combining the data from the naïve and all timepoints. The color scale represents the most stable gene (yellow) to the least stable gene (blue), (n ≥ 27 rats). (**C**) Ranking of the nine HKGs according to Ct value, Ct value standard deviation, gene stability from the five algorithms including naïve and injured tissues, and protein expression level in naïve tissues. The color scale represents the most (yellow) to least (blue) suitable endogenous reference gene.

**Figure 5 mps-06-00022-f005:**
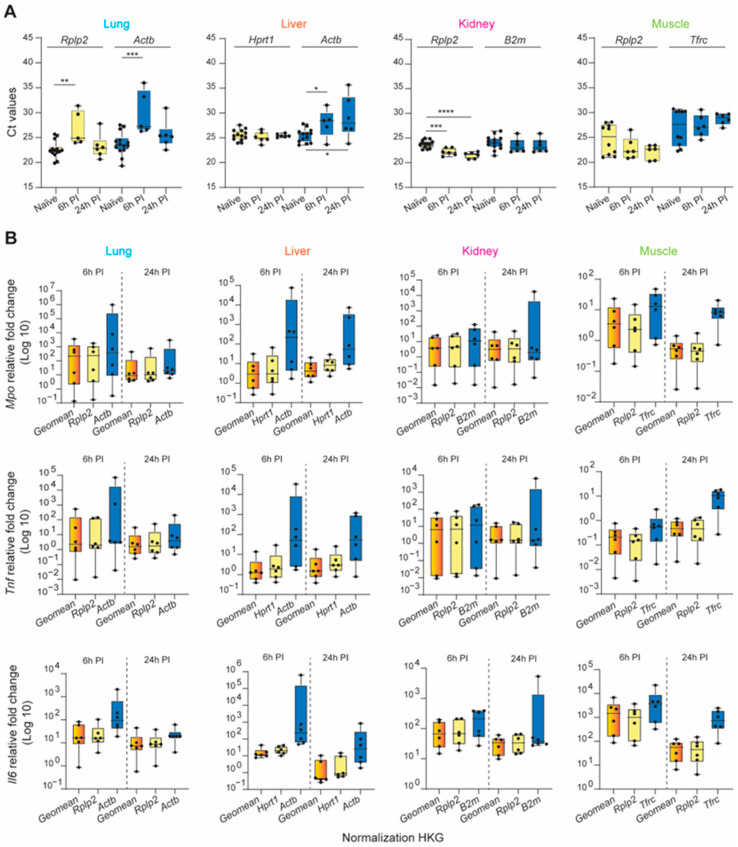
Validation of selected HKGs to normalized inflammatory biomarker gene expression across traumatic experimental conditions in the lung, liver, kidney, and muscle. (**A**) Average of the Ct values obtained for the most (yellow bars) and least (blue bars) stable HKGs within different timepoints: naïve, 6 h post-injury (6 h PI), and 24 h (24 h PI). Statistical significance was defined as * *p* ≤ 0.05; ** *p* ≤ 0.01, *** *p* ≤ 0.001, and **** *p* ≤ 0.0001. (**B**) Average of relative fold-change expression of inflammatory biomarkers, myeloperoxidase (*Mpo*), tumor necrosis factor (*Tnf*), and Interleukin 6 (*Il6*), by either the geometric mean of the top three highly stable HKGs (orange bars), and the most (yellow bars) and least (blue bars) stable HKGs for the lung, liver, kidney, and muscle tissues; n = 6 rats for each experimental condition. Error bars represent standard deviations.

**Table 1 mps-06-00022-t001:** Summary of the nine HKGs evaluated in this study.

Symbol	Name	Physiological Functions	RefSeq Accession No.	Bio-Rad Assay ID	Efficiency	r^2^	Amplicon Length (bp)
*Actb*	Beta-actin	Cell motility and cytoskeletal maintenance [43,44,45]	NM_031144	qRnoCID0056984	97	0.9987	74
*B2m*	Beta-2-microglobulin	Assembly and surface expression of MHC class I molecules [46]	N/A	qRnoCED0056999	95	0.9998	106
*Gapdh*	Glyceraldehyde-3-phosphate dehydrogenase	Glycolysis [47]; transcription activation; initiation of apoptosis [48,49]; vesicle trafficking [50]	NM_017008	qRnoCID0057018	96	0.9998	115
*Hprt1*	Hypoxanthine-guanine phosphoribosyltransferase	Purine nucleotide generation [51]	NM_012583	qRnoCED0057020	98	0.9989	79
*Pgk1*	Phosphoglycerate kinase 1	Phosphoprotein glycolysis [52]	NM_053291	qRnoCED0002588	98	0.9993	78
*Rplp0*	60S acidic ribosomal protein Stalk Subunit P0	Elongation step of protein synthesis [53,54]	NM_022402	qRnoCED0005242	100	1	97
*Rplp2*	Ribosomal Protein Lateral Stalk Subunit P2	Elongation step of protein synthesis [53,54]	N/A	qRnoCED0015635	89	0.9911	60
*Tbp*	TATA-box-binding protein	Activation of eukaryotic genes [55]	NM_001004198	qRnoCID0057007	95	0.9985	107
*Tfrc*	Transferrin receptor	Regulating stellate cell activation [56]	NM_022712	qRnoCID0003700	96	0.9998	66

## Data Availability

The authors confirm that the data supporting the findings of this study are available within the manuscript and the Appendix A. Raw data presented in this study are available on request from the corresponding author and will be made available to others in the private and public sector as soon as appropriate agreements covering such transfer can be executed pursuant to USUHS policies regarding data sharing.

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
