# Peer review of "Systematic Identification of the Optimal Housekeeping Genes for Accurate Transcriptomic and Proteomic Profiling of Tissues following Complex Traumatic Injury"

_mps, 2023, doi:10.3390/mps6020022_

Round 1
Reviewer 1 Report
The manuscript from Dragon et al. investigated the expression of housekeeping genes in an animal model of extremity trauma.
Major comments:
- How were the nine “common” housekeeping genes investigated in this study selected? There are no references to support that all the nine genes are the ones most used in gene and protein expression injury and post-traumatic studies. Are different housekeeping genes used in different species and/or different injury models? Additional references and information are needed.
- Methods: Why only male Sprague Dawley rats were used?
- Methods: Authors wrote: “The specifics of the nine candidate HKGs tested, and the primer/probe sets used in this study are presented in Table 1.” According to the Bio-Rad website, the assays listed in Table 1 seem to be PCR primers only, not primers and probe, which aligns with the use of Sybr Green and melting curves. Please correct.
- Methods: While Authors stated that the cDNA was generated with 600 ng of total RNA, it is not clear how much cDNA was used for each RT-qPCR reaction. Were dilutions made from the cDNA? A more detailed description is needed.
- Figure 1: For clarity, include in the caption which samples were used for the results shown in Figure 1. Were they injury samples or non-injury samples?
- Figure 1: How much volume of PCR product was loaded in the gel from each sample?
- Figure 2: Please clarify how the 2-(delta delta CT) was calculated when the housekeeping genes are the target genes in this study.
- Figure 2C: Include the markers for the protein ladder in the figure and the meaning of the numbers “1, 2, 3” in the figure panel.
- Figure 2C: A positive control is needed for each Western blot, since some bands are very weak and there is no loading control because the housekeeping proteins are the targets. As it is, these results are not interpretable and the conclusion of most and least expressed (Figure 2D) is not acceptable.
- Figure 3: There is clearly a lot of variation on the CT values of injured tissues (large standard deviations of the mean). How is that considered in the calculation of the stability ranking?
- Add references and/or a rationale for the selection of myeloperoxidase (Mpo), tumor necrosis factor (Tnf) and interleukin 6 (Il6) genes as validation targets.
Minor comments:
- Small typos in the manuscript need to be corrected, such as on page 2: “rapidly and reliably quantitate m large number of mRNA molecules”.
Author Response
We thank the reviewers and editor for their inputs and comments. We have addressed all the comments, and have formatted the manuscript as per instructions from the editorial board. Edits are made using track changes in this revisited manuscript.
Please find below the responses to the comments.
Comments from Reviewer 1:
How were the nine “common” housekeeping genes investigated in this study selected? There are no references to support that all the nine genes are the ones most used in gene and protein expression injury and post-traumatic studies. Are different housekeeping genes used in different species and/or different injury models? Additional references and information are needed.
Response: We thank the reviewer for this very helpful feedback. For consistency and clarity, we have add the following text to the Methods section of the manuscript. “We selected nine common reference housekeeping genes based on a targeted literature search conducted using the PubMed database (2013-2023) involving qPCR data normalization studies (Table S1) across diverse spectrums of tissues, disease models, and animal species. For the most common references genes reported in the literature, it is not always clear on what basis these control genes were chosen and for those used for gene expression normalization, which HKGs were the most appropriate. It’s important to note that very few of these reports involved critical validation studies”
Methods: Why only male Sprague Dawley rats were used?
Response: The Sprague-Dawley male rat is used extensively as an animal model to characterize combat-related extremity trauma, in particular injury patterns involving blast exposure. We have been funded and have extensively published studies involving quantitative biomarker/gene signature analyses in response to various forms of traumatic injury using the Sprague Dawley rat.
Methods: Authors wrote: “The specifics of the nine candidate HKGs tested, and the primer/probe sets used in this study are presented in Table 1.” According to the Bio-Rad website, the assays listed in Table 1 seem to be PCR primers only, not primers and probe, which aligns with the use of Sybr Green and melting curves. Please correct.
Response: Bio-Rads qPCR probe fluorescently labeled oligonucleotide assays are designed to enable multiplexing, which we did not use. Instead, we utilized Bio-Rad’s pre-validated PCR primer pairs in which each assay “PCR primer pair” was dried in a single well in a 384-well plate. For the SYBR Green, we used Bio-Rad SsoAdvanced Universal SYBR Green Supermix (cat# 172-5275) with ROX normalization. We have made this clear in the Methods section and have included a copy of the validation reports supplied by Bio-Rad to avoid further confusion.
Methods: While Authors stated that the cDNA was generated with 600 ng of total RNA, it is not clear how much cDNA was used for each RT-qPCR reaction. Were dilutions made from the cDNA? A more detailed description is needed.
Response: We thank the reviewer for pointing out this omission. We used 10 ng of cDNA per qPCR reaction. This has been indicated in the revised manuscript.
Figure 1: For clarity, include in the caption which samples were used for the results shown in Figure 1. Were they injury samples or non-injury samples?
Response: For the melt curve analyses, we utilized 4 biological replicates from injured lung 6h post-injury. We have updated the figure legend to reflect these clarifications.
Figure 1: How much volume of PCR product was loaded in the gel from each sample?
Response: We thank the reviewer for pointing out this omission. Each sample was loaded with 7.5 µl of PCR product per lane. This has been indicated in the revised manuscript.
Figure 2: Please clarify how the 2-(delta delta CT) was calculated when the housekeeping genes are the target genes in this study.
Response: Figure 2A illustrates Ct values not 2-(delta delta Ct) values. Selected HKGs and the 2-(delta delta Ct) method were used to calculate the relative fold gene expression of MPO, IL6 and TNF (Figure 4).
Figure 2C: Include the markers for the protein ladder in the figure and the meaning of the numbers “1, 2, 3” in the figure panel.
Response: We thank the reviewer for this comment. The referenced numbers relate to sample (biological) replicates not sample repeats. This has been indicated in the revised Figure 2C.
Figure 2C: A positive control is needed for each Western blot, since some bands are very weak and there is no loading control because the housekeeping proteins are the targets. As it is, these results are not interpretable and the conclusion of most and least expressed (Figure 2D) is not acceptable.
Response: We thank the reviewer for this very helpful feedback. Positive controls were not run, however equal amounts of protein per lane was run and confirmed by Ponceau S solution intensity staining (Figure S2A) for total protein normalization. Ponceau S resulted in the same sensitivity of protein band detection for each tissue biological replicate. The sensitivity of protein detection in conjunction with consistent intensity of HKG bands amongst tissue biological replicate samples (proofing the targeted HKG by band comparison) gives credibility to this work. The methods and results have been revised to include these points.
Figure 3: There is clearly a lot of variation on the CT values of injured tissues (large standard deviations of the mean). How is that considered in the calculation of the stability ranking?
Response: The statistically-based algorithms used in this study, use either coefficient of variance (CV) and/or regression parameter for determining stability ranking. Amongst these, there is some high correlation, but also some discrepancies between algorithms. For example, Bestkeeper ranks HKGs based on the standard deviation (SD) and CV expressed as a percentage of the cycle threshold (Ct) level, whereas NormFinder accounts for both intra- and inter-group variations. Moreover, RefFinder assigns an appropriate weight to an individual gene and calculates the geometric mean of their weights for the overall final ranking based on the complied rankings from geNorm, Normfinder, BestKeeper, and the comparative ΔCt. Our results support the concept that more than one computational methodology is necessary for accurate determining stability ranking.
Add references and/or a rationale for the selection of myeloperoxidase (Mpo), tumor necrosis factor (Tnf) and interleukin 6 (Il6) genes as validation targets.
Response: We thank the reviewer for this very helpful feedback. A systemic inflammatory response (SIRS) can results from various types of acute trauma, ischemia-reperfusion injury, and infection. This response involves immune cell activation and a complex network of proinflammatory cytokines, which may induce multiple organ failure when uncontrolled. IL-6, TNFg and myeloperoxidase are just a few of the key factors/targets which play key roles in this response. We have cited some applicable references discussing the role of these targets in regulating the acute immune inflammatory response.
Minor comments:
Small typos in the manuscript need to be corrected, such as on page 2: “rapidly and reliably quantitate m large number of mRNA molecules”.
Response: We thank the reviewer for pointing out this overlooked typo. We have made the suggested edit.
Reviewer 2 Report
The paper "Systematic Identification of the Optimal Housekeeping Genes for Accurate Transcriptomic and Proteomic Profiling of Tissues Following Complex Traumatic Injury" by Dragon and colleagues describes their effort in identifying housekeeping genes in injured tissues (lung, kidney and muscle). The article is well written and provides experimental validation, but it is currently very limited in two aspects: 1) generalization (it is not a protocol, but rather an analysis of a specific histological context) and 2) de novo HK genes detection (it is currently flawed by having manually selected a few candidate genes). Below, my specific comments.
- The article doesn't currently constitute the basis for an extendable Protocol, as the choice of journal would imply. The authors should spend some words on how their study could be generalized for other contexts. For example, I particularly liked the comparison with naive tissue (Figure 3A), which could be applied for example on the comparison between healthy and tumor tissues (e.g. by comparing data from GTEX and TCGA). In other words, the authors should rewrite the article in a form that is generalizable for housekeeping gene finding in other contexts.
- The main problem with the article is that it tests preselected candidate housekeeping genes, but it is currently unable to determine new candidate housekeeping genes. This can be problematic for tissues and histological scenarios where none of the predefined HK genes are stable expression-wise. The authors must provide a rationale for pre-selecting candidate housekeeping genes, for example by providing the top candidate HK genes based on public data, explaining the requirements of a good candidate, namely 1) high expression mean and 2) low expression variance.
Author Response
We thank the reviewers and editor for their inputs and comments. We have addressed all the comments, and have formatted the manuscript as per instructions from the editorial board. Edits are made using track changes in this revisited manuscript.
Please find below the responses to the comments.
Author's Reply to the Review Report (Reviewer 2)
Top of Form
The paper "Systematic Identification of the Optimal Housekeeping Genes for Accurate Transcriptomic and Proteomic Profiling of Tissues Following Complex Traumatic Injury" by Dragon and colleagues describes their effort in identifying housekeeping genes in injured tissues (lung, kidney and muscle). The article is well written and provides experimental validation, but it is currently very limited in two aspects: 1) generalization (it is not a protocol, but rather an analysis of a specific histological context) and 2) de novo HK genes detection (it is currently flawed by having manually selected a few candidate genes). Below, my specific comments.
The article doesn't currently constitute the basis for an extendable Protocol, as the choice of journal would imply. The authors should spend some words on how their study could be generalized for other contexts. For example, I particularly liked the comparison with naive tissue (Figure 3A), which could be applied for example on the comparison between healthy and tumor tissues (e.g. by comparing data from GTEX and TCGA). In other words, the authors should rewrite the article in a form that is generalizable for housekeeping gene finding in other contexts.
Response: We thank the reviewer for this comment. The intent was to publish a focused article identifying stable common housekeeping genes suitable for quantification and normalization of qRT-PCR data following acute trauma injury and to share the validation process and outcomes of what we have learned with the scientific community. Our findings highlight a critical problem in previous investigations, and are generalized enough for other contexts. Using the datasets in this study, we strongly feel we have provided an “extended protocol” which provide insight into a generalized validation strategy and may be helpful for others to accurately assess gene expression patterns with high degree of confidence multiple species and under diverse conditions, treatments, developmental processes and diseases. This applies both for Western blot and RT-qPCR.
Expectations for real-time RT-PCR technique impose an extremely carefully made research, which may be a reference for many other techniques. Thousands of studies are based on qPCR data while in fact only a small number of de novo studies are being developed. To obtain reliable results it is necessary to carry out the process of normalization with reference genes and to interpret them rationally.
Response: We agree with the reviewers overall assessment.
The main problem with the article is that it tests preselected candidate housekeeping genes, but it is currently unable to determine new candidate housekeeping genes. This can be problematic for tissues and histological scenarios where none of the predefined HK genes are stable expression-wise. The authors must provide a rationale for pre-selecting candidate housekeeping genes, for example by providing the top candidate HK genes based on public data, explaining the requirements of a good candidate, namely 1) high expression mean and 2) low expression variance.
Response: We thank the reviewer for this comment. Large-scale gene expression profiling of hundreds of candidate HKGs to find the most stable tissue-specific HKGs to conduct gene analyses for a DoD funded combat-related extremity trauma grant was beyond the focus of the scope of work and breadth of our research intentions. We selected nine common reference housekeeping genes based on a targeted literature searches conducted using the PubMed database (2013-2023) involving qPCR studies (Table 1) across diverse spectrums of tissues, disease models, and animal species. For the most common references genes reported in the literature, it is not always clear on what basis these control genes were chosen and for those used normalization, which were the most appropriate. It’s important to note that very few of these reports involved critical validation studies, in particular information related to expression levels (Ct values, variance, and translational). We feel that the findings from our study add critical value for reporting biological meaningful results.
Round 2
Reviewer 1 Report
The response to the question: “Methods: Why only male Sprague Dawley rats were used?” was not satisfactory. This Reviewer is very familiar with Sprague-Dawley rats as a model for traumatic injury. My question was regarding the use of only male rats, instead of both male and female rats as a model. Less arrogance is expected from Authors when answering to a Reviewer’s questions.
The response to the question: “Figure 2: Please clarify how the 2-(delta delta CT) was calculated when the housekeeping genes are the target genes in this study.” was also not satisfactory. While it is clear that Figure 2A illustrates Ct values, Figure 2B heatmap shows a Delta Ct (ΔCt), which was not clarified. Is this Delta Ct the variation of two Ct values? Which ones?
Author Response
- The response to the question: “Methods: Why only male Sprague Dawley rats were used?” was not satisfactory. This Reviewer is very familiar with Sprague-Dawley rats as a model for traumatic injury. My question was regarding the use of only male rats, instead of both male and female rats as a model. Less arrogance is expected from Authors when answering to a Reviewer’s question.
Response: We appreciate the reviewer's experience with the Sprague-Dawley rat. Our initial response was not at all intended to be arrogant, and we apologize for giving that perception. Our initial response was correct except that we should have included the word "male" again in the 2nd sentence - as noted, our team has extensive experience with a blast/polytrauma model in these animals, and all of that prior work has been done utilizing male rodents. The reasons for this are for physiologic consistency (e.g., avoiding potential hormone-related biomarker differences due to reproductive cycles) and ensuring test rodents do not become pregnant - although our team is of course meticulous in our research practices, the veterinary technicians responsible for cage maintenance, daily feeding, etc. do not work directly for us and might be less so in terms of gender confirmation and caging. Finally, in terms of clinical relevance, we believe that our model is appropriate as fewer than 3% of severely combat-injured service members from the recent conflicts were female (Ref 1). While we believe our findings would likely be similar and relevant to combat-injured women as well as men, our model is thus appropriate for at least the 97% of male combat casualties.
1. Hylden C, Johnson AE, Rivera JC. Comparison of female and male casualty cohorts from conflicts in Iraq and Afghanistan. US Army Medical Department journal. 2015 Apr 1:70-6.
- The response to the question: “Figure 2: Please clarify how the 2-(delta delta CT) was calculated when the housekeeping genes are the target genes in this study.” was also not satisfactory. While it is clear that Figure 2A illustrates Ct values, Figure 2B heatmap shows a Delta Ct (ΔCt), which was not clarified. Is this Delta Ct the variation of two Ct values? Which ones?
Response: We understand the author’s statement , insight and apologize for not clearly addressing the concern on how the stability ranking was conducted using the ΔCt algorithm to develop the heatmap illustrated in Figure 2B. The Delta Ct algorithm uses the standard deviation (SD) of the Ct values to rank the stability of all reference genes, and the reference gene with the lowest SD value shows the most stable performance (Ref 2).
2. Zhang J, Xie W, Yu X, Zhang Z, Zhao Y, Wang N, Wang Y. Selection of Suitable Reference Genes for RT-qPCR Gene Expression Analysis in Siberian Wild Rye (Elymus sibiricus) under Different Experimental Conditions. Genes (Basel). 2019 Jun 13;10(6):451. doi: 10.3390/genes10060451. PMID: 31200580; PMCID: PMC6627066.
Reviewer 2 Report
The authors answered all my comments, improving the paper in the process. As a result, I believe the study is now more powerful and extended on more contexts where housekeeping genes are involved.
Author Response
Thank you for the positive feedback.